



**The precipitation variability of wet and dry season at the interannual**
**and interdecadal scales over eastern China (1901–2016): The impacts**
**of the Pacific Ocean**
Tao Gao[1, 4, 5], Fuqiang Cao [2, 3*], Li Dan[3], Ming Li[2], Xiang Gong[5], and Junjie Zhan[6]
[1] State Key Laboratory of Numerical Modeling for Atmospheric Sciences and
Geophysical Fluid Dynamics, Institute of Atmospheric Physics, Chinese Academy of
Sciences, Beijing 100029, China
[2] School of geosciences, Shanxi Normal University, Linfen 041000, China
[3] CAS Key Laboratory of Regional Climate-Environment Research for Temperate
East Asia, Institute of Atmospheric Physics, Chinese Academy of Sciences, Beijing,
China
[4] College of Urban Construction, Heze University, Heze 274000, China
[5] School of Mathematics and Physics, Qingdao University of Science and Technology,
Qingdao 266061, China
[6] Shunyi Meteorological Service, Beijing, 101300, China
*Corresponding author address: Dr. Fuqiang Cao, 1 Gongyuan Road, Linfen 041000,
P. R. China.
Email: wq2006126@126.com





**Abstract:** The spatiotemporal variability of rainfall in dry (October- March) and wet (April-September) seasons over eastern China is examined based on gridded rainfall dataset from University of East Angela Climatic Research Unit during 1901–2016. Principal component analysis is employed to identify the dominant variability modes, wavelet coherence is utilized to investigate the spectral characteristics of leading modes of precipitation and their coherences with the large-scale modes of climate variability, and Bayesian dynamical linear model is adopted to quantify the time-varying correlations between climate variability modes and rainfall in dry and wet seasons. Results show that first and second principal components (PCs) account for 34.2% (16.1%) and 13.4% (13.9%) of variance in dry (wet) season, and their changes are roughly coincident with phase shifts of the El Niño-Southern Oscillation (ENSO) in both seasons. The anomalous moisture fluxes responsible for the occurrences of precipitation events in eastern China are asymmetry during high and light rainfall years in dry (wet) season. ENSO has a 4- to 8-year signal of the statistically positive (negative) association with rainfall during dry (wet) season in eastern China. The statistically significant positive (negative) associations between Pacific Decadal Oscillation (PDO) and precipitation are found with 9- to 15-year (4- to 7-year) signal. The impacts of PDO on rainfall in eastern China exhibit multiple time scales as compared to ENSO episodes, while PDO triggers a stronger effect on precipitation in wet season than dry season. The interannual and interdecadal variations in rainfall over eastern China are substantially modulated by drivers originated from Pacific Ocean, the finding has meaningful implications for regional hydrologic predictability and water resources management.

**Keywords:** Precipitation over eastern China; Principal component analysis; Wavelet spectral analysis; Bayesian dynamical linear model



## 1. Introduction


As a densely populated area with lots of industrial and agricultural activities, eastern
China is frequently affected by the catastrophic floods and droughts derived from
variability of precipitation events (Liu et al., 2015; Huang et al., 2017; Yang et al.,
2017; Luo and Lau, 2018; Ge et al., 2019). For example, intense rainfall in southern
China resulted in disastrous floods over the lower reach of Yangtze River basin (YRB)
in 1991, 1996, 1998 and 1999. Seriously deficient precipitation in northern China
caused a severe drought of 226 days without stream discharge over the Yellow River
basin (Qian and Zhou, 2014; Xu et al., 2015; Zhang and Zhou, 2015). It is therefore of
great importance to investigate the rainfall variability in eastern China and its
associated physical mechanisms.
Many studies pointed out that the variations in rainfall in eastern China are strongly
influenced by East Asian monsoon, which is closely related to the sea surface
temperature (SST) anomalies over the Pacific Ocean (Wang and Zhou, 2005; Huang
et al., 2017; Yang et al., 2017). At the interannual scale, heavy rainfall events often
occur over southern China during El Niño episodes (e.g., Zhang et al., 1996; Wang et
al., 2000; He et al., 2017). At the interdecadal scale, the variations in precipitation
events over eastern China are remarkably impacted by tropical Pacific SST and
western Pacific subtropical high (WPSH, Chang et al., 2000a; Zhu et al., 2011; Li et
al., 2019). Moreover, SST anomalies over the tropical Indian Ocean and tropical
eastern Pacific also account for the shifts of the positive-negative-positive rainfall
patterns over eastern China via their influences on WPSH (Chang et al., 2000b; Hu et





al., 2018). Thus, a better understanding of interannual and interdecadal changes
stemming from the variability of air-sea interaction over the Pacific Ocean is crucial
to the interpretation for the variations in rainfall over eastern China.

74        The El Niño-Southern Oscillation (ENSO) is a strong air-sea coupled mode at the

interannual scale over the tropics, it is also the important source of interannual
variability of the global climate system (Webster et al., 1998). ENSO significantly
impacts rainfall over eastern China by means of the atmospheric teleconnections (e.g.,
Wang et al., 2008; Jin et al., 2016; Liu et al., 2016; Sun et al., 2017; Gao et al., 2017).
Wang et al. (2000) proposed that the key system of Pacific-East Asian teleconnection
responsible for linkages between ENSO and precipitation anomalies over eastern
China is an anomalous low-level anticyclone located over the western North Pacific
(WNP), this is induced by local air-sea interactions and large-scale equatorial heating
anomalies. Wu et al. (2003) further argued that the similar positive correlation
between springtime rainfall over the mid-lower reaches of YRB and ENSO is linked
to the evolution of ENSO-related seasonal rainfall anomalies over East Asia.
Moreover, the summertime rainfall over the YRB and to its south is expected to be
strengthened (weakened) during El Niño (La Niña) years. Huang and Wu (1989)
documented that the drought in northern and southern China as well as flood over
central China are associated with the developing stage of warm ENSO events, and the
reversed relationship is seen in decaying stage of the warm events. These patterns of
rainfall in eastern China may also be related to strong convective activities in the
Philippines, with the effects from western Pacific warm pool through shifting the





WPSH northward (Huang and Sun, 1992; Jin et al., 2016). The latest research
suggested that the patterns of seasonal rainfall anomaly in eastern China are
modulated by the different types of La Niña decay, these are attributed to the
responses of large-scale circulation anomalies induced by different types of La Niña
episodes (Chen et al., 2019).
At the interdecadal scale, northern China experienced dry and wet alternations,
with above-normal rainfall around the 1950s and severe droughts around the 1970s
and 1980s. While the YRB and southern China suffered apparent shifts of
precipitation patterns in the 1970s and 1990s (Zhu et al., 2015). A growing body of
studies indicated that these shifts of rainfall distribution over eastern China are caused
by the changes in Pacific decadal oscillation (PDO) phases. Yang and Lau (2004)
reported a close relationship between the positive PDO and decreasing trends of
summertime rainfall events over eastern China. Based on surface wetness indices, Ma
(2007) further pointed out an anti-correlation between rainfall in northern China and
PDO phases, suggesting more droughts over northern China during positive phase of
PDO, and vice versa. The strengthened (weakened) precipitation over the Huang-Huai
(Yangtze) River basin from 2000 to 2008 compared to those during 1979-1999 is
triggered by the transition from warm to cold phase of the PDO around the 2000s,
which is attributed to the weakened westerly winds and warming over the Lake Baikal
induced by negative PDO after 2000s (Zhu et al., 2011). The possible modulation of
the PDO on the East Asian summer monsoon (EASM) and East Asian winter
monsoon (EAWM), which are associated with summer and winter rainfall changes in





eastern China, respectively, has also documented in previous studies (e.g., Yu, 2013;
Chen et al., 2013). Zhou et al. (2013) pointed out an anti-correlation between the PDO
and EASM since 1950s, and negative phases of the PDO correspond to a stronger
EASM with more precipitation events over northern China. A much stronger EASM
tends to appear after a weak EAWM in positive phases of the PDO than that in
negative phases of the PDO (Chen et al., 2013). Existing studies also reported the
similar relationship between positive phase of the PDO and drier conditions in
northern China, and revealed that a warm phase of PDO in the 1976/1977 resulted in a
weakened EASM associated with aridity over northern China in the 1980s and 1990s
(Qian and Zhou, 2014; Zhu et al., 2015; Yang et al., 2017; Gao and Wang, 2017).
Furthermore, the relationship between interdecadal variability of rainfall patterns over
eastern China and phase transitions of PDO is also identified and verified by coupled
climate model simulations (e.g., Li et al., 2010; Yu et al., 2015).
The above analyses show that most previous studies focused on the impacts of
ENSO and PDO on the variations in seasonal rainfall over eastern China. However,
the main rainy season in China, particularly in eastern China, does not follow
climatological seasonal boundaries. Usage of boreal standard seasons may therefore
unavoidably break the natural rainy distribution at the temporal scale, affecting the
robustness of the analytical results (Zhai et al., 2005). Up to now, the issue on whether
the ENSO and PDO can contribute to the interannual and interdecadal rainfall
variability in major rainy seasons over eastern China is still unclear. In this study, we
utilize April–September as the wet half year (wet season) and October–March as the



dry half year (dry season), respectively, to examine the effects of ENSO and PDO on
the precipitation variability at the space-time scale, since the rainfall in eastern China
is principally concentrated during April–September (Bao 1987; Domroes and Peng
1988; Zhai et al., 2005). Data and methods are described in section 2. The results are
provided in section 3. Section 4 presents the discussion and conclusions.
**2. Data and Methods**
**2.1 Data**
A dataset of daily accumulated rainfall amount at 756 meteorological stations during
1960-2015 across China is employed in this study. This dataset is developed at
Climate Data Center of the National Meteorological Center of the China
Meteorological Administration (http://cdc.cma.gov.cn/dataSetDetailed.do), including
almost all the first and second class national climatological stations. The accurate
quality control procedures are conducted to check the temporal inhomogeneity and
missing values, and screen the related stations in the following analyses, meaning that
the stations having too many missing rainfall values are dropped. For example, a year
is considered as the missing year if there exists more than 10% missing days, and a
station with less than 5% missing years is retained. After these procedures, 436
stations meet these criteria and are retained in the subsequent analyses. Another
rainfall dataset is a Global land monthly precipitation dataset from University of East
Angela Climatic Research Unit (CRU), which has a high resolution of $0.5° \times 0.5°$
over land from 1901 to 2016. The CRU data covers a longer period as compared to
observed counterpart, therefore, it is more suitable for examining multi-decadal



variability. More information about this dataset is referred to Harris et al. (2014).
We select monthly global circulation variables from National Centers for
Environmental Prediction/National Center for Atmospheric Research (NCEP/NCAR)
Reanalysis data (Kalnay et al., 1996). SST data are obtained from the Hadley Centre,
Met Office (Rayner et al., 2003). ENSO index is obtained from the Climate Prediction
Center of NOAA
(http://origin.cpc.ncep.noaa.gov/products/analysis_monitoring/ensostuff/detrend.nino
34.ascii.txt). The PDO index is extracted from the Earth System Research Laboratory
of NOAA (http://www. esrl.noaa.gov/psd/data/correlation/pdo.data/).
**2.2 Method**
**2.2.1 Principle component analysis**
The gridded CRU precipitation dataset is subjected to the principle component
analysis (PCA), which is a widely utilized method to extract the dominant temporal
and spatial modes of the variability based on mutually correlated dataset. The leading
principal component (PC) explains the most of variance, with the second PC
decreases thereafter. Moreover, the leading PCs can reduce dimension of the original
dataset, because they capture the most of variance. The detailed description of the
PCA refers to Hannachi et al. (2007). To identify the effects of climate variability
modes on variations in rainfall over eastern China, the correlations between the
leading PCs and climate variability modes are calculated to understand the
telecommunications. The composited maps of the atmospheric variables are analyzed
to examine the physical mechanisms responsible for the rainfall variability, based on




the high and light 25th percentile values of the daily rainfall in wet and dry seasons,
respectively.

### 2.2.2 Wavelet coherence

The wavelet coherence is a widely used technique, based on how coherent the
cross-wavelet transform is in time frequency space. It can preferably access the
detailed relationships between two time series with different time periods and
disparate frequency ranges (e.g., Grinsted et al., 2004; Coulibaly and Burn, 2005).
Given two particular time series $x_n$ and $y_n$, the wavelet coherence of them can be
expressed as
$$W^{XY} = W^X W^{Y*} \qquad (1)$$

where $*$ represents their complex conjunction. Correspondingly, the cross-wavelet
power can be expressed as $\left| W^{XY} \right|$. And complex argument arg $\left( W^{xy} \right)$ is considered as
local relative phases between the time series $x_n$ and $y_n$, which are applicative in both
frequency and time domains. The wavelet coherence of the time series can be defined
according to Torrence and Webster (1999).
$$R_n^2(s) = \frac{\left| S(s^{-1} W_n^{XY}(s)) \right|^2}{S(s^{-1} \left| W_n^X(s) \right|^2) \cdot S\left( s^{-1} \left| W_n^Y(s) \right|^2 \right)} \qquad (2)$$

where $S$ is the smoothing operator, which is further written as,
$$S_{time}(W) = S_{scale}(S_{time}(W_n(s))) \qquad (3)$$

where $S_{scale}$ and $S_{time}$ denote the smoothing along wavelet scale axis and time,
respectively. It is natural to design the smoothing operator so that it has a similar
footprint as the wavelet.



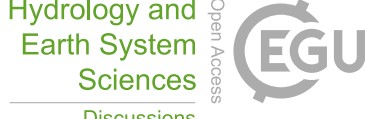

The related codes for the wavelet coherence used in the present study can be freely
downloaded    from    http://www.pol.ac.uk/home/research/waveletcoherence/.    The
wavelet coherence is used to examine the correlation between ENSO/PDO and
rainfall over eastern China.

**2.2.3 Bayesian dynamic linear model**

The increases in amplitude of the SST anomaly patterns over the Pacific Ocean in the
context of global warming trigger non-stationarity changes in regional rainfall (Wang
et al., 2013; Krishnaswamy et al., 2015; Rajagopalan and Zagona, 2016). The
Bayesian dynamic linear model (BDLM) is employed to analyze the non-stationarity
and epochal fluctuations between the climate variability modes and rainfall in eastern
China. The description of BDLM model as follows,

$$\begin{cases} y_t = \alpha_t + x_t \beta_t + v_t, & v_t \sim N(0, V_t) \\ \alpha_t = \alpha_{t-1} + \omega_{\alpha,t}, & \omega_{\alpha,t} \sim N(0, W_{\alpha,t}) \\ \beta_t = \beta_{t-1} + \omega_{\beta,t}, & \omega_{\beta,t} \sim N(0, W_{\beta,t}) \end{cases} \qquad (4)$$

where  $y_t$  is the leading PCs of rainfall over eastern China,  $x_t$  is the covariate
(climate variability modes, i.e., ENSO and PDO), and  $\alpha_t$  and  $\beta_t$  are the dynamic
intercept and slope coefficients at time  $t$ .  $\omega_t$  is the corresponding evaluation error
and  $W_t$  is the corresponding scalar greater than zero.
Unlike traditional linear regression methods that cannot characterize the
time-varying relationship, BDLM can model and understand the non-stationarity in
the relationships between large-scale modes of climate variability and regional
precipitation with time. This method has been used to model monsoonal precipitation
variability in India and China, and shows better performance and more interesting





insights than the traditional regression method (Krishnaswamy et al., 2015; Gao et al.,
2017). For the BDLM, the regression coefficient varies with time compared to the
traditional regression, in which the coefficient remains fixed.
**3. Results**
**3.1 Comparison between observed and CRU rainfall datasets**
The variations in monthly and annual rainfall over eastern China based on both the
observed stations and CRU gridded points from 1960 to 2015 are illustrated in Fig. 1.
The monthly mean precipitation is shown in dashed lines and the climatological
average is depicted in solid red lines (Fig. 1a, b). It can be seen in Fig. 1 that the
climatological variability of observed rainfall along with months is quite similar to
CRU gridded dataset. The slight discrepancy is that the annual mean rainfall is larger
and smaller than 80 mm for CRU and observed datasets, respectively. The
climatological rainfall is greater (lesser) than annual mean value from April to
September (October to March), consistent with the periods of wet (dry) season (half
year) selected in this study. These changes confirm that there is reasonable to
categorize wet and dry seasons in conjunction with the variations in rainfall over
eastern China. We further compare the time series of mean rainfall between
observation and CRU datasets during wet and dry seasons (Fig. 1c, d), which suggest
a strong level of similarity between observed and CRU datasets. High spatial
similarity of the observed and CRU datasets during dry (Fig. 2a, c) and wet (Fig. 2b, d)
seasons indicates that the spatial patterns from these two datasets are also consistent.
In addition, the spectral analysis is also performed using the mean rainfall series of



the two datasets (not shown) and the similar results are also obtained. Those indicate
that the rainfall variability for CRU dataset coincides with observations over eastern
China. We use CRU dataset since it covers a much longer period and is therefore
more suitable to examine the interdecadal variability. We present the following
analyses in wet and dry seasons, respectively, to provide a concise result.
**3.2 Dry season**
The two leading PCs explain 34.22% and 13.44% of the total variance, they together
capture around 50% variance. Fig. 3 depicts the time series of first PC that is flipped
for convenient comparison, which is suggestive of a well correspond with the spatial
mean rainfall. The first two eigenvectors, including spatial components and
corresponding PCs, are shown in Fig. 4. The spatial pattern of the first eigenvector
exhibits similar magnitudes and signs, indicating that the dominant pattern is coherent
in eastern China, especially over southern China and coastal regions (Fig. 4a), this
may be related to the propagation of the EAWM into mainland China. The second
eigenvector displays a southeast-southwest dipole over southern China, this feature is
coincident with the location and movement of the EASM (Ding et al., 2009). The time
series of PCs also show considerable temporal changes with time, which are discussed
in the spectral analysis.
Fig. 5 shows the correlation maps of climate variables and PC1 and PC2. Note that
the signs of the PCs are flipped to ensure that the correlations are directly inferred as
rainfall variability over eastern China. The correlation between PC1 and SSTs
displays strong positive coefficients over the equatorial tropical Pacific and North





Pacific. While the negative connections are mainly found over the South China Sea
(SCS) and central-east Pacific, where it is featured by a La Niña SST pattern (Fig. 5a).
This indicates that when the eastern Pacific is colder as it is the case in La Niña
episodes, the strengthened convections may occur over southern China and adjacent
areas, leading to increased precipitation events, and vice versa in El Niño events. The
pattern of correlation with SLP is inconsistent with ones for SSTs, the significant
positive correlations are principally seen over the South Pacific and some tropical
regions immediate close to the Indian and Pacific oceans (Fig. 5b). Whereas, some
significant positive coefficients are located over the East China Sea, this may enhance
the southeastern wind anomalies that transport more water vapor fluxes into southern
China, providing conducive environmental backgrounds of forming more rainfall
events. Considering correlations with the geopotential heights at the 500 hPa (Fig. 5c),
the significant negative coefficients over the tropical central-east Pacific suggest a
weakened EAWM. When the EAWM weakens, the strengthened cold and dry air
intrudes into southern China and converges with warm and wet air from the oceans,
facilitating the occurrence of convective activities and heavy precipitation events
(Huang et al., 2018).

284         Correlation of SSTs with PC2 is reminiscent of the El Niño pattern, even though it

is not evident (Fig. 5d), an indication suggests that El Niño episode yields a dipole
pattern of the rainfall over southern China during dry season. The correlations with
SLP exhibiting positive coefficients are mainly distributed in the North Pacific and
Siberia, while the negative coefficients are principally situated over the equatorial



Pacific and Indian Oceans (Fig. 5e). Correlation coefficient between PC2 and 500 hPa
is relatively smaller and barely remarkable (Fig. 5f). Those imply that larger portion
of the variability induced by climate variables occurs in the first mode.
Composited analyses of anomalous water vapor fluxes and divergence based on
highest 75th and lightest 25th percentile rainfall values, respectively, during dry
season are shown in Fig. 6. Considering the 25th percentile conditions, an anomalous
anticyclone is found over the WNP, while one branch of anomalous moisture fluxes to
the southern flank is transported eastward to eastern Pacific, meanwhile, another
branch is transported westward to Indian Ocean (Fig. 6a). As a result, the divergence
appears over eastern China, which is not suitable for the occurrence of precipitation
events. The adverse phenomena are found for the 75th percentile events (Fig. 6b). The
westward transportation of anomalous water vapor fluxes is prominent over the
equatorial pacific, converging with the eastward transportation of moisture flux
anomalies from Indian Ocean over the SCS. Then the converged moisture fluxes are
transported northward, forming an anomalous cyclone over the WNP. The anomalous
water vapor fluxes over northern and western flanks of the WNP are transported into
eastern China, and anomalous terrestrial water vapor fluxes from Eurasia are also
transported into study domain. Those patterns provide favorable environmental
background and sufficient moisture supply for the formation of the convergence,
which is conducive to the occurrences of heavy rainfall events.
The wavelet coherence is performed on the PCs with large-scale ocean-atmosphere
circulation patterns to investigate the temporal variability of leading modes of rainfall





(Fig. 7). The local and global spectrums of PC1 indicate spectral peaks in the 1- to
4-year band and 6- to 10-year band further, which seems to be active during recent
decades (Fig. 7a). For PC2, the 1- to 4-year band is active before the middle part of
the twentieth century, while the 5- to 7-year band is concentrated in recent decades
(Fig. 7b). ENSO index (Niño3.4) exhibits a significant peak of 2- to 7-year period and
a relatively weaker peak of 8- to 16-year period (Fig. 7c). Fig. 7e displays that ENSO
has a positive association with rainfall from 1900 to 1930, with a 4- to 8-year signal.
There is also a positive relationship from 1980 to 2010, with an 1- to 6-year signal.
These suggest that ENSO has a statistically positive impact on precipitation over
eastern China in dry season. Wavelet filtering of the PC1 in the 4- to 8-year period
with ENSO being coherent (Fig. 7c) is also made and illustrated in Fig. 3 as the solid
line. PDO has a statistically positive connection with rainfall from 1940 to 1970, with
a 7- to 8-year signal. While a negative association is seen from 1980 to 2000, with an
8- to 9-year signal (Fig. 7f). Particularly, the PDO is closely related to precipitation
over eastern China.
**3.3 Wet season**
The total variance captured by first two PCs is about 30%, with PC1 and PC2
explaining 16.06% and 13.93%, respectively during wet season. These are smaller
than total variances explained by two leading PCs of rainfall during dry season. The
spatial mean precipitation is also captured by first PC (Fig. 8), which is flipped for
easily comparing with spatial pattern. The solid line indicates the decadal smoother of
first PC, and will be discussed later. While the low frequency of temporal variability





is seen in Fig. 8. The spatial components and corresponding PCs of first two
eigenvectors are shown in Fig. 9. A north-south dipole pattern is found for the first
eigenvector, with strong negative values located over southern China (Fig. 9a), which
has a close correlation with the variability of spatial mean precipitation (Fig. 8). This
rainfall pattern is also associated with the location and propagation of the EASM (Jin
et al., 2016). In wet season, the northward advance of the EASM circulations is
followed by three major rainy seasons sequentially: from May to mid-June, early
summer rainy season occurs in southern China. Then the mei-yu season presents over
the Yangtze-Huai river basins. The late summer rainy season ultimately forms over
northern China (Ding and Chan, 2005). Correspondingly, multiple synoptic and
climatological systems contribute to the occurrence of these rainfall events (Gao et al.,
2016; Luo et al., 2016). The second eigenvector exhibits the magnitudes of the
coherent signs in eastern China, with the peaks over the mid-lower reaches of YRB
(Fig. 9b). Moreover, the first two PCs display considerable temporal changes (Fig. 9c,
d) that are described in the discussion of spectral analysis.

348       The correlation map of PC1 with SSTs shows the strong positive coefficients over

the North Pacific and western tropical Pacific (Fig. 10a), while some statistically
negative correlations are distributed in the WNP. The positive correlations with SLP
exhibiting statistical significance are seen over the eastern Pacific, and the negative
values are found over the WNP and oceans to the eastern Australia (Fig. 10b). This is
roughly an opposite correlation pattern of SLP in comparison with dry season (Fig. 5b
and 10b). For 500 hPa, the positive correlations are mainly located over the WNP,





with positive values principally situated over the equatorial western Pacific, which are
weaker as compared to the correlations in dry season. The correlation between SSTs
and PC2 exhibits evident spatial features (Fig. 10d). Statistically significant negative
coefficients are principally discovered over the eastern Pacific, reminiscent of the La
Niña episode, this is suggestive of the La Niña telecommunication mechanisms
responsible for the rainfall over eastern China during wet season. Note that
statistically significant positive coefficients are mainly distributed over the northern
Indian Ocean, resembling the Indian Ocean basin mode. To response the basin-wide
warming of Indian Ocean, the strengthened convective heating in the tropical Indian
Ocean will drive the Kelvin-wave-like eastern anomalies to the east. Then, the
anticyclonic shear of the Kelvin-wave-like easterlies may drive the boundary layer
divergence over the WNP by Ekman pumping, and therefore suppresses convection
there. These suppressed convections simulate an anomalous anticyclone to the west.
Ultimately, the anomalous anticyclone in the tropical WNP intensifies rainfall over
eastern China (Li et al., 2017; Cao et al., 2020). The correlation of PC2 with SLP is
much weaker compared to that of PC1, with significant negative coefficients located
over far WNP (Fig. 10e). There also exists a weaker correlation with 500 hPa in
comparison with PC1, and negative values mainly situate over the WNP (Fig. 10f).

373       Composited maps of moisture fluxes and divergence in high and light precipitation

years during wet season are illustrated in Fig. 11. Unlike the anomalous changes in
dry season (Fig. 6), the anomalous westward transportation of water vapor fluxes is
found over the equatorial Pacific for the lightest 25th percentile precipitation events,



while the water vapor anomalies that are transported from Indian Ocean into eastern
China are not apparent (Fig. 11a). However, anomalous moisture fluxes are
transported northeastward passing eastern China, and fail to from convergence over
eastern China, which is not suitable for the occurrences of rainfall events. Fig. 11a
shows that eastern China is principally dominated by divergence during light rainfall
years. For the highest 75th percentile precipitation events, an anomalous cyclone
appears over the WNP, even though it is relatively weak (Fig. 11b). The water vapor
anomalies originated from WNP converge with those from Eurasia over eastern China,
as illustrated in Fig. 11b. Most of the eastern China is dominated by convergence,
providing inductive environmental backgrounds of the occurrences of heavy rainfall
events. In addition, the anticyclone and cyclone are seen over the Indian Ocean during
light and high rainfall years, respectively, which is generally consistent with the
Indian Ocean capacitor effects on the Indo–western Pacific climate in summer (Xie et
al., 2009).
The local and global spectrum of PC1 suggests the spectral peaks in the 1- to 5-year
and 6- to 10-year bands, as well as 16- to 32-year band further, these periods are
likely more active during recent decades (Fig. 12a). On the other hand, the PC2 shows
2- to 5-year and 5- to 8-year bands, as well as 16- to 24-year band. The first period
seems to be active in recent decades, and second and third periods are active from
1920 to 1980 (Fig. 12b). The ENSO index exhibits remarkable peaks of the 3- to
7-year period, which is active after 1950s (Fig. 12c). ENSO events have a statistically
negative relationship with rainfall over eastern China in wet season, with a 4- to





8-year signal, while other signals are not evident enough, even though they occur
intermittently during the entire twentieth century (Fig. 12e). These suggest that the
modulation of ENSO on wet season precipitation is mainly concentrated at the
interannual scale, consistent with those in dry season. This also coincides with the
interannual band of the wavelet filtering of the PC1 (Fig. 8). Fig. 12f shows that PDO
events have statistically significant positive associations with wet season rainfall from
1920 to 1940, with a 9- to 15-year signal. The significant negative connection with
rainfall exhibits a 4- to 7-year signal from 1930 to 1950. It can be seen from Fig. 7f
and Fig. 12f that PDO events have a stronger influence on rainfall in wet season than
that in dry season.

409        The changing connections between leading modes of precipitation and large-scale

modes of climate variability with time are accessed by BDLM (Fig. 13). We display
the results that have discernable changes along with time, and ignore the results
without discernable variations. The intercept from BDLM of PC1 and ENSO exhibits
a slight increase from 1920 to 1960, then turns into a decrease condition and
experiences zero value around the 1980s (Fig. 13a), suggesting that ENSO triggers a
negative (positive) impact before (after) the 1980s, and the influences of ENSO
become strengthened during recent decades. The intercept of PC2 and ENSO shows
negative values, and is gradually decreasing with time, which indicates that the
impacts of ENSO on PC2 are weakening during the entire century (Fig. 13b).
Considering the effects of PDO, the positive connection between PDO and PC1
exhibits a decrease until 1980s, then the impacts of PDO on rainfall over eastern



China are strengthening in recent decades (Fig. 13c). However, almost the opposite
phenomenon is found for the connection between PC2 and PDO (Fig. 13d). The
negative intercept is getting close to zero with time before 1980s, suggesting that the
impact of PDO on PC1 is decreasing during this period. Then the positive connection
of PC2 and PDO become strengthened after 2000s, indicating that the effect of PDO
on PC2 is enhanced after this period. These results are important applications on the
predictability of the rainfall events over eastern China based on the ENSO and PDO
(Gao et al., 2017), since the ENSO and PDO has impacted the predictability of early
summer monsoon precipitation in south China with the changes in connections
between climate variability modes and rainfall (Chan and Zhou, 2005).
**4. Discussion and conclusions**
Space-time variability of rainfall during dry and wet seasons over eastern China is
examined by utilizing PCA, wavelet coherence and BDLM, based on the CRU
gridded and observed rainfall datasets. In the overlapping period of 1960-2015, these
two rainfall datasets are consistent in their temporal and spatial patterns in both
seasons over eastern China. While the CRU gridded data has a much longer period
(1901-2016) and is more suitable to analyze the interdecadal variability of rainfall.
The PCs exhibit notably temporal changes at the interannual and interdecadal
scales. In dry season, the first and second eigenvectors account for 34.2% and 13.4%
of variance, they exhibit coherent and dipole patterns of rainfall over southeastern
China and southern China, respectively, which are generally coincident with the shifts
of ENSO phases. Particularly, the strengthened rainfall over southeastern China is



associated with the La Niña episodes, and the dipole pattern of precipitation in
southern China occurs during El Niño years. Moreover, the variations in rainfall over
eastern China during dry season are also affected by the intensity of EAWM and the
patterns of SLP. In wet season, first and second eigenvectors show dipole and
coherence of rainfall patterns, respectively, which are roughly contrary to that in dry
season. And the two leading PCs account for 16.1% and 13.9% of variance,
respectively. The circulations responsible for the changes in rainfall over eastern
China are also generally opposite to those during dry season.
Composited analyses illustrate the southeastward and southwestward
transportations of moisture flux anomalies from southern portion of eastern China,
and there is no convergence occurred over study region for 25th percentile rainfall
events during dry season. In the years with highest (75th percentile) rainfall events,
the anomalous moisture fluxes from equatorial Pacific and Indian Ocean are
transported into eastern China through SCS, leading to the convergence with the
anomalous water vapor fluxes from WNP and Eurasia in eastern China, providing
sufficient moisture supply and environmental backgrounds for the occurrences of
precipitation events. In wet season, the anomalous variations in moisture fluxes are
different with that during dry season. For the lightest rainfall years, the water vapor
anomalies that are transported from equatorial Pacific pass through eastern China, this
northeastward transportation of water vapor anomalies fails to form a convergence in
study region. Thus, most of the eastern China is consequently dominated by the
divergence. However, the opposite phenomena are found for the 75th percentile
events, the water vapor anomalies from WNP converge with the anomalous moisture
fluxes from Eurasia, they are transported southwestward into eastern China, resulting
in heavy precipitation events. Note that the anticyclone and cyclone in Indian Ocean
also play an important role to the occurrences of rainfall events over eastern China in
addition to the forcing factors originated from Pacific Ocean (Xie et al., 2009; Li et al.,

470  2017).

ENSO has a statistically positive (negative) association with rainfall during dry
(wet) season in eastern China, with a 4- to 8-year signal. The impacts of ENSO on
rainfall are principally concentrated at the interannual scale in both dry and wet
seasons. PDO has a statistically positive (negative) relationship with rainfall in both
seasons, exhibiting a 7- to 8-year (8- to 9-year) signal in dry season. And the
statistically significant positive (negative) associations between PDO and
precipitation over eastern China is seen with 9- to 15-year (4- to 7-year) signal. In
short, the effects of PDO on rainfall show multiple time scales compared to these of
ENSO. Moreover, the PDO triggers a stronger impact on precipitation over eastern
China in wet season than dry season. Previous studies have revealed that PDO has a
significant effect on the movement of rainbelt over eastern China during the rainy
seasons, which influence the spatial distribution of rainfall events (i.e., southern flood
and northern drought) (Li et al., 2010; Gao et al., 2017). Our findings further confirm
those phenomena in eastern China at the interdecadal scale.
The analyses using BDLM suggest that there exists no significant time-varying
relationship between large-scale modes of climate variability and rainfall over eastern



China in dry season. In wet season, the intercept of ENSO and PC2 gradually
decreases with time, suggesting that the influences of ENSO on PC2 are gradually
weakening in the entire century. The effect of PDO on PC1 is decreasing before 1980s,
then shifts into positive connection after 2000s. The insights of spatiotemporal
variability of rainfall over eastern China at different time scales, and the temporal
variability of the strengths between climate variability modes (ENSO and PDO) and
rainfall will be of great importance for developing skillful precipitation forecasting
model. Moreover, BDLM provides a flexible regression method to incorporate the
predictors with varying strengths, the model parameters are therefore estimated
dynamically at each time, which enable to capture the time-varying predictors. The
results in this study can also be adopted to develop seasonal precipitation forecasting
models. Particularly, the asymmetry of the rainfall over eastern China and ENSO
teleconnections in dry and wet seasons indicate the different underlying causes during
El Niño and La Niña episodes, which can potentially improve the forecasting skills,
these phenomena are also true for different phases of PDO episodes. The physical and
human infrastructures over eastern China have suffered from severe floods and
droughts, therefore, the skillful hydroclimate projections of space–time variability of
rainfall will facilitate policy makers to develop the effective mitigation strategies.




**Author contributions.**

Gao T and Cao F designed all the experiments. Gao T and Cao F conducted all the

experiments and analyzed the results. All the authors contributed to the preparation of

the English editing.

**Competing interests.**

The authors declare that they have no conflict of interest.

**Acknowledgments**

This study is jointly supported by Natural Science Foundation and Sci-tech

development project of Shandong Province (No. ZR2018MD014; J18KA210), Key

research and development plan of Shandong province in 2019 (No. 2019GGX105021),

Project funded by China Postdoctoral Science Foundation (No. 119100582H;

1191005830), and Project of National Natural Science Foundation of China (No.

41630532).



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





**Figure captions**

Figure 1. Annual climatological rainfall at all stations (STN) and grid (CRU) points shown as grey lines and their mean in a solid red line, (a) observation and (b) CRU. Seasonal mean precipitation anomalies from observation (black) and CRU (blue), (c) dry season and (d) wet season.

Figure 2. Spatial distribution of seasonal mean precipitation (mm/month) during 1960-2015 over eastern China from observation and CRU datasets, (a) and (c) are for dry season; (b) and (d) are for wet season.

Figure 3. Standardized time series of all dry season precipitation over eastern China as shown in red dashed line, the black dots denote flipped PC1 and the blue lines denote the decadal features of dry season precipitation.

Figure 4. (a) The first and (b) second EOFs for the rainfall in dry season. (c) The first and (d) second principal components (PCs) correspond to these EOFs from the rainfall in dry season. Both time series are normalized with respect to the corresponding standard deviations.

Figure 5. Correlation coefficients in dry season. (a) sea surface temperature and PC1, (b) mean sea level pressure with PC1, (c) geopotential height at the 500 hPa and PC1, (c) sea surface temperature and PC2, (e) mean sea level pressure with PC2 and (f) geopotential height at the 500 hPa with PC2. Hatching denotes the regions with statistical significance at the 95% confidence level. Black rectangle denotes the eastern China.

Figure 6. Vertically integrated water vapor anomalies (vector) and water vapor flux divergence (shading) composited from the lightest 25th (a) and highest 75th (b) percentile rainfall events in dry season. The water vapor flux unit is kg m$^{-1}$ s$^{-1}$ for and the water vapor flux divergence is kg m$^{-2}$ s$^{-1}$. Green rectangle denotes the eastern China.

Figure 7. Wavelet spectra for dry season. (a) PC1, (b) PC2, (c) Niño3.4 index, (d) PDO index, (e) wavelet spectral coherence of PC1 and Niño3.4, and (f) wavelet spectral coherence of PC2 and PDO. The global spectra are shown on the right side of





the time varying wavelet spectra and, the black lines denote the statistical significance
at the 95% confidence level.
Figure 8. Standardized time series of all wet season precipitation over eastern China
as shown in red dashed line, the black dots denote flipped PC1 and the blue lines
denote the decadal features of wet season precipitation.
Figure 9. (a) The first and (b) second EOFs for the rainfall in wet season. (c) The first
and (d) second principal components (PCs) correspond to these EOFs from the
rainfall in wet season. Both time series are normalized with respect to the
corresponding standard deviations
Figure 10. Correlation coefficients in wet season. (a) sea surface temperature and PC1,
(b) mean sea level pressure with PC1, (c) geopotential height at the 500 hPa and PC1,
(c) sea surface temperature and PC2, (e) mean sea level pressure with PC2 and (f)
geopotential height at the 500 hPa with PC2. Hatching denotes the regions with
statistical significance at the 95% confidence level. Black rectangle denotes the
eastern China.
Figure 11. Vertically integrated water vapor anomalies (vector) and water vapor flux
divergence (shading) composited from the lightest 25th (a) and highest 75th (b)
percentile rainfall events in wet season. The water vapor flux unit is kg m$^{-1}$ s$^{-1}$ for and
the water vapor flux divergence is kg m$^{-2}$ s$^{-1}$. Black rectangle denotes the eastern
China.
Figure 12. Wavelet spectra for wet season. (a) PC1, (b) PC2, (c) Niño3.4 index, (d)
PDO index, (e) wavelet spectral coherence of PC1 and Niño3.4, and (f) wavelet
spectral coherence of PC2 and PDO. The global spectra are shown on the right side of
the time varying wavelet spectra and, the black lines denote the statistical significance
at the 95% confidence level.
Figure 13. Changes in the relationships between rainfall and ENSO/PDO over time
during 1901-2015. Black solid lines denote the estimated time-varying slopes, along
with 25th and 75th percentile credible interval lines (red dotted lines) from the
Bayesian dynamic linear model analysis.


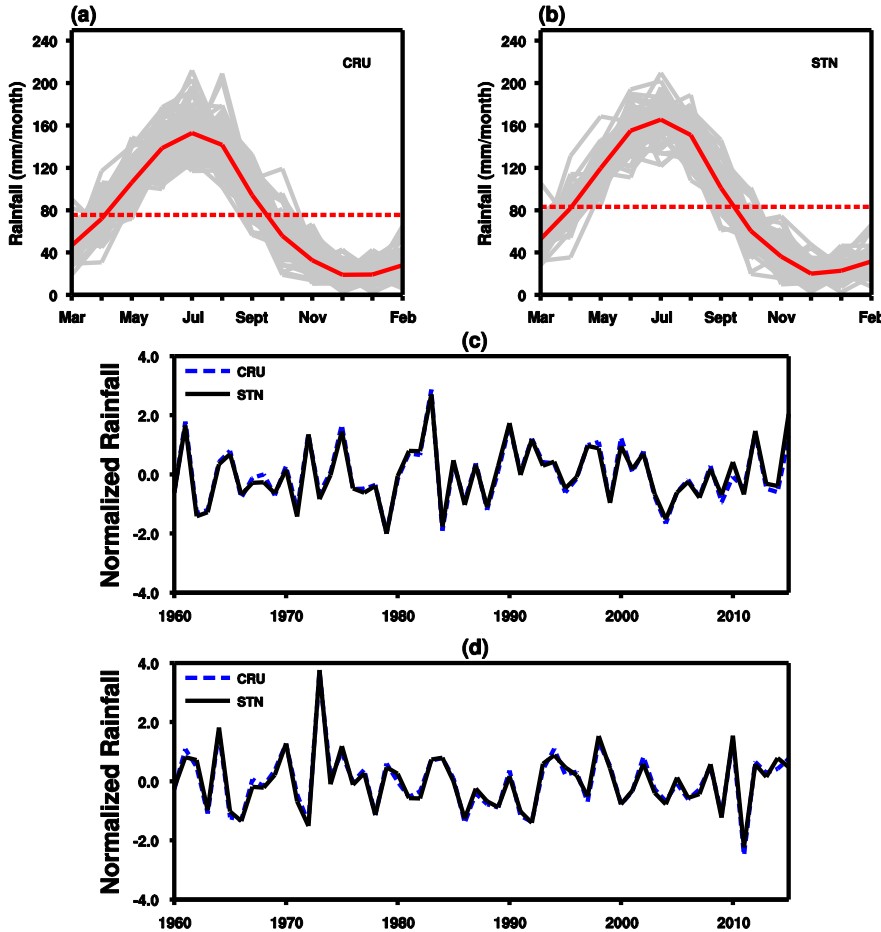


Figure 1. Annual climatological rainfall at all stations (STN) and grid (CRU) points

shown as grey lines and their mean in a solid red line, (a) observation and (b) CRU.

Seasonal mean precipitation anomalies from observation (black) and CRU (blue), (c)

dry season and (d) wet season.




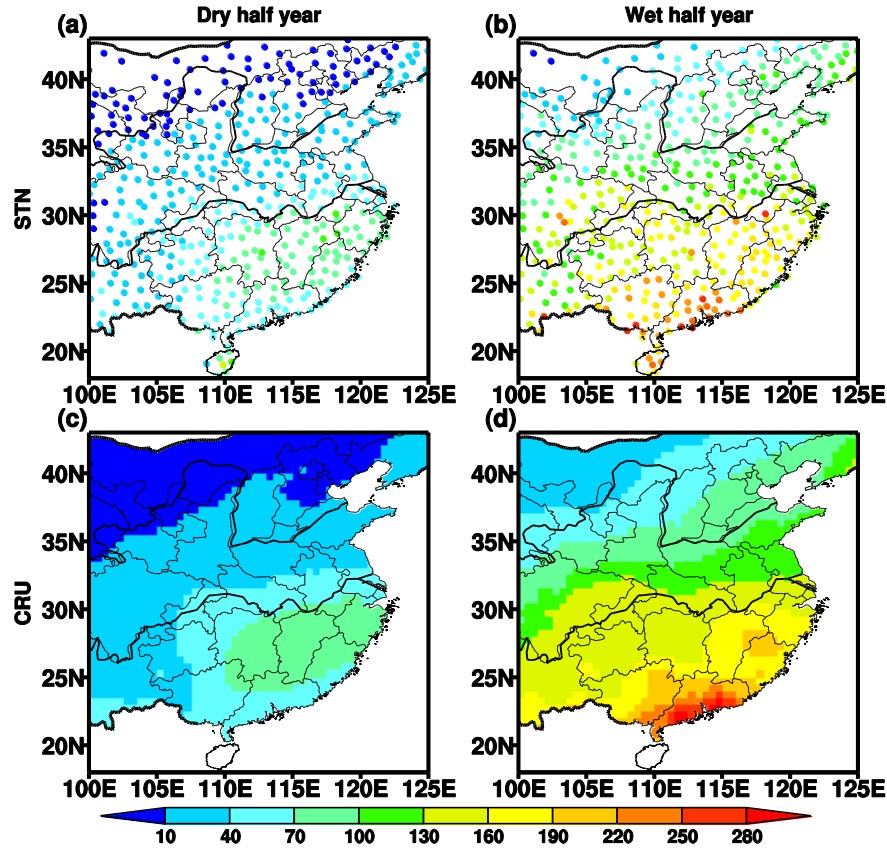


Figure 2. Spatial distribution of seasonal mean precipitation (mm/month) during

1960-2015 over eastern China from observation and CRU datasets, (a) and (c) are for

dry season; (b) and (d) are for wet season.





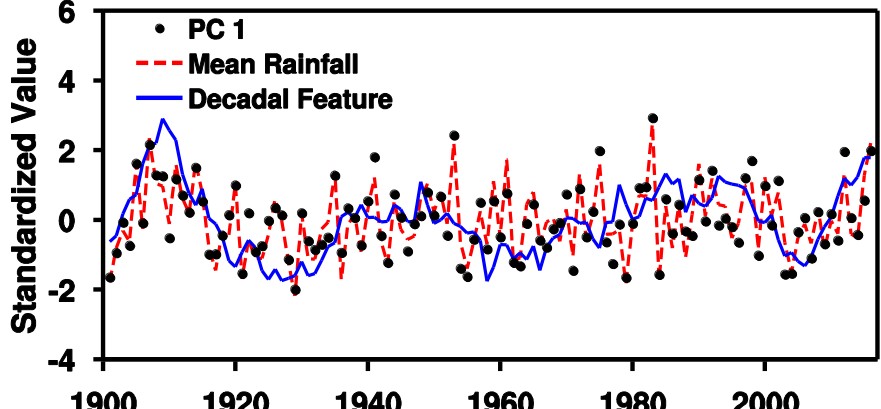


Figure 3. Standardized time series of all dry season precipitation over eastern China as

shown in red dashed line, the black dots denote flipped PC1 and the blue lines denote

the decadal features of dry season precipitation.






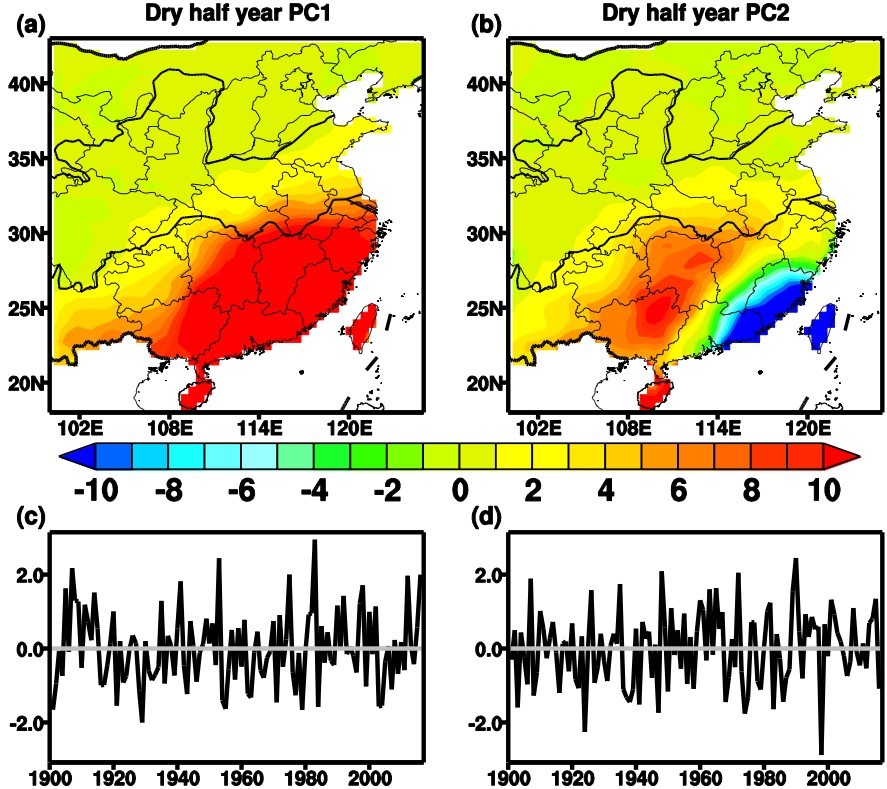

Figure 4. (a) The first and (b) second EOFs for the rainfall in dry season. (c) The first
and (d) second principal components (PCs) correspond to these EOFs from the
rainfall in dry season. Both time series are normalized with respect to the
corresponding standard deviations.


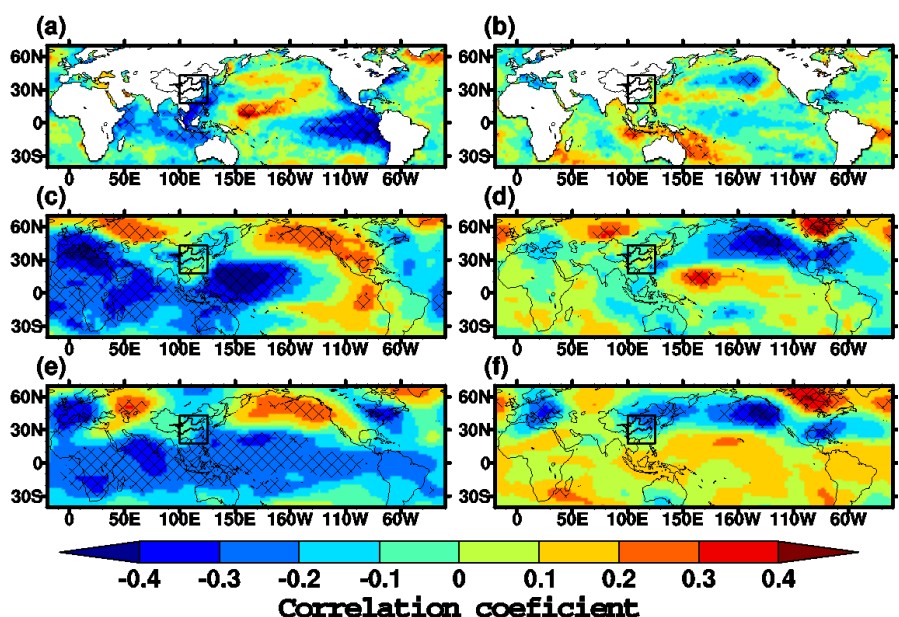


Figure 5. Correlation coefficients in dry season. (a) sea surface temperature and PC1,

(b) mean sea level pressure with PC1, (c) geopotential height at the 500 hPa and PC1,

(c) sea surface temperature and PC2, (e) mean sea level pressure with PC2 and (f)

geopotential height at the 500 hPa with PC2. Hatching denotes the regions with

statistical significance at the 95% confidence level. Black rectangle denotes the

eastern China.

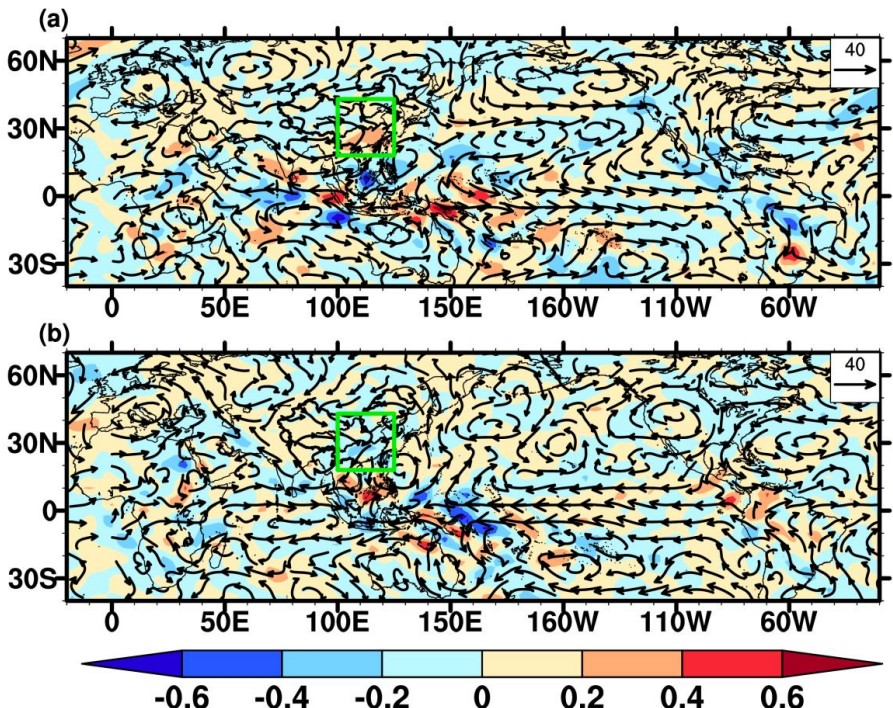

788

Figure 6. Vertically integrated water vapor anomalies (vector) and water vapor flux

divergence (shading) composited from the lightest 25th (a) and highest 75th (b)

percentile rainfall events in dry season. The water vapor flux unit is kg m$^{-1}$ s$^{-1}$ for and

the water vapor flux divergence is kg m$^{-2}$ s$^{-1}$. Green rectangle denotes the eastern

China.



795

796

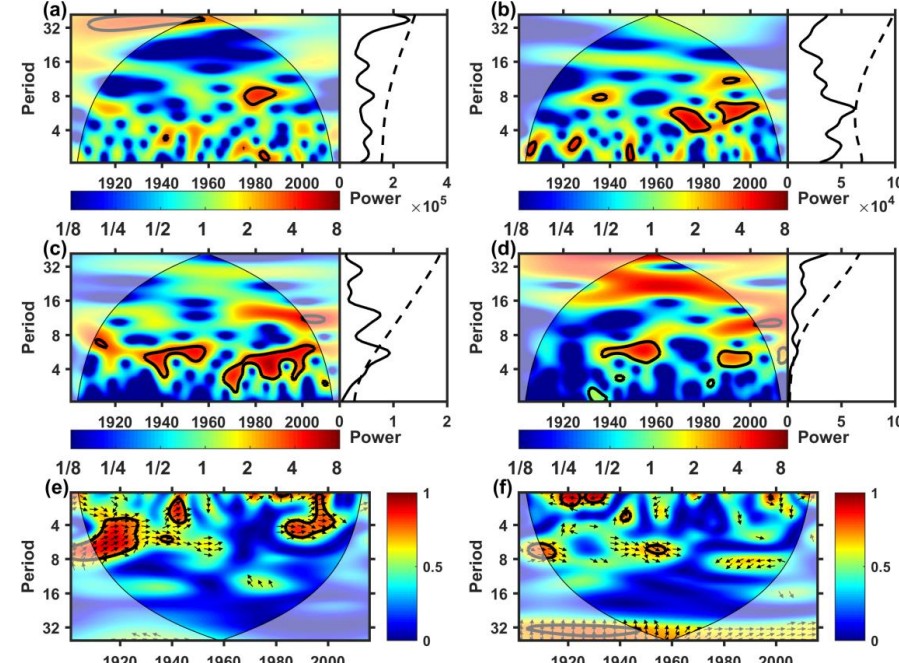

797

Figure 7. Wavelet spectra for dry season. (a) PC1, (b) PC2, (c) Niño3.4 index, (d)

PDO index, (e) wavelet spectral coherence of PC1 and Niño3.4, and (f) wavelet

spectral coherence of PC2 and PDO. The global spectra are shown on the right side of

the time varying wavelet spectra and, the black lines denote the statistical significance

at the 95% confidence level.






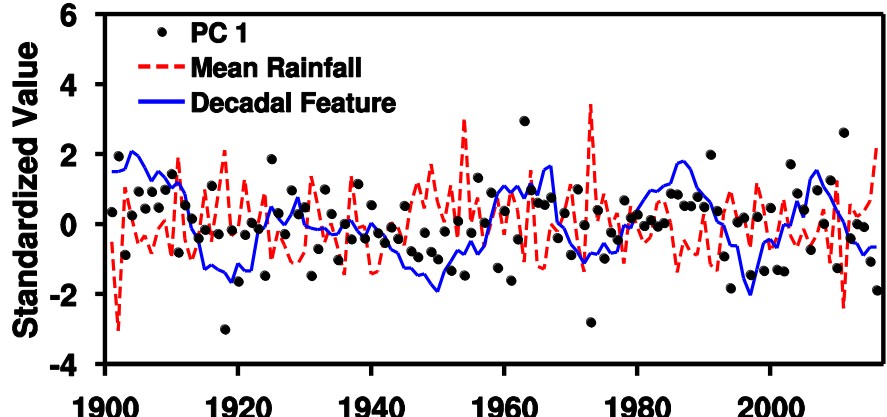


Figure 8. Standardized time series of all wet season precipitation over eastern China
as shown in red dashed line, the black dots denote flipped PC1 and the blue lines
denote the decadal features of wet season precipitation.







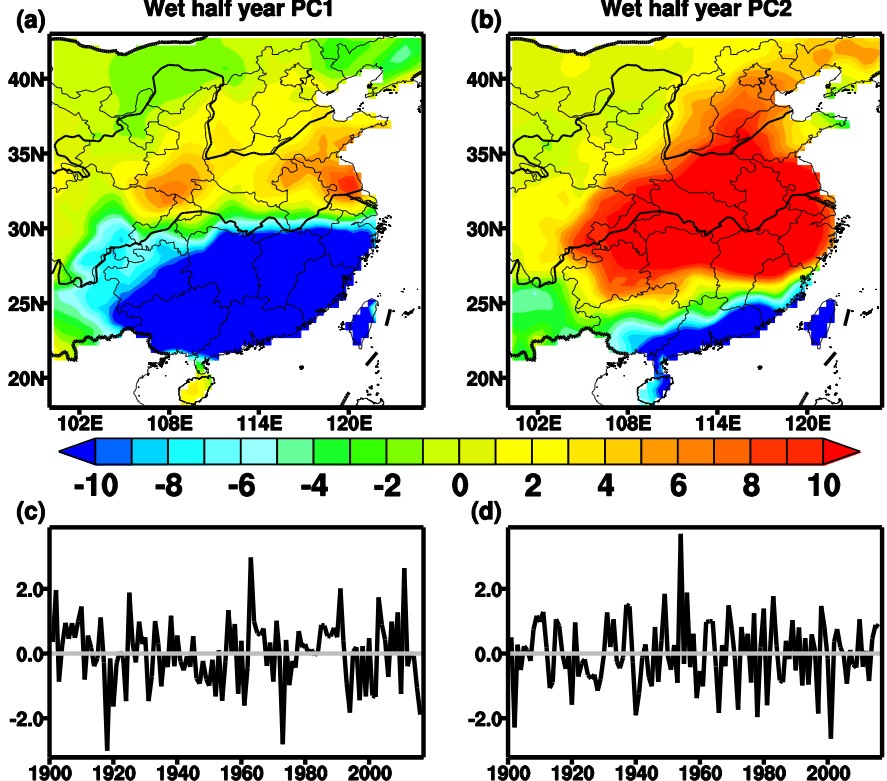

Figure 9. (a) The first and (b) second EOFs for the rainfall in wet season. (c) The first
and (d) second principal components (PCs) correspond to these EOFs from the
rainfall in wet season. Both time series are normalized with respect to the
corresponding standard deviations.




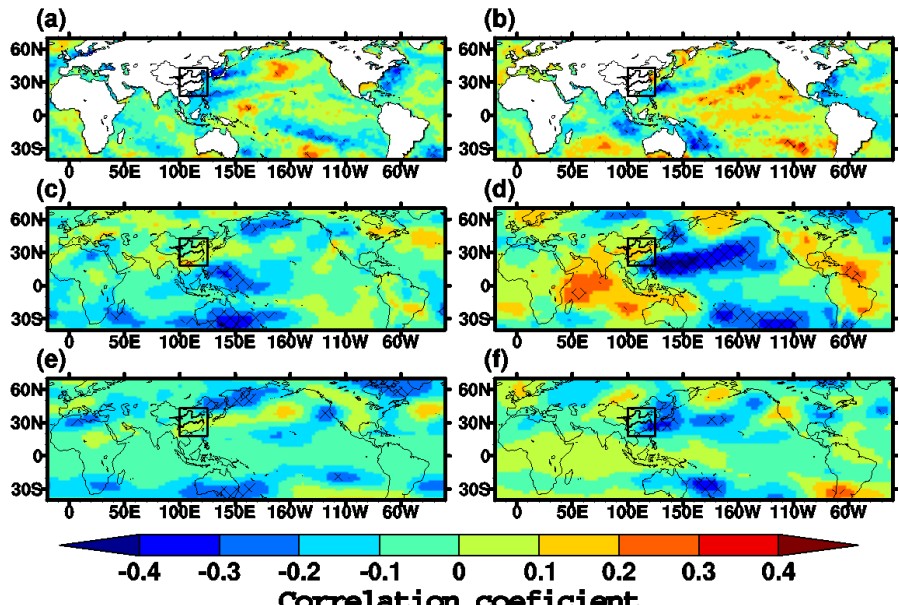


Figure 10. Correlation coefficients in wet season. (a) sea surface temperature and PC1,

(b) mean sea level pressure with PC1, (c) geopotential height at the 500 hPa and PC1,

(c) sea surface temperature and PC2, (e) mean sea level pressure with PC2 and (f)

geopotential height at the 500 hPa with PC2. Hatching denotes the regions with

statistical significance at the 95% confidence level. Black rectangle denotes the

eastern China.

828

829

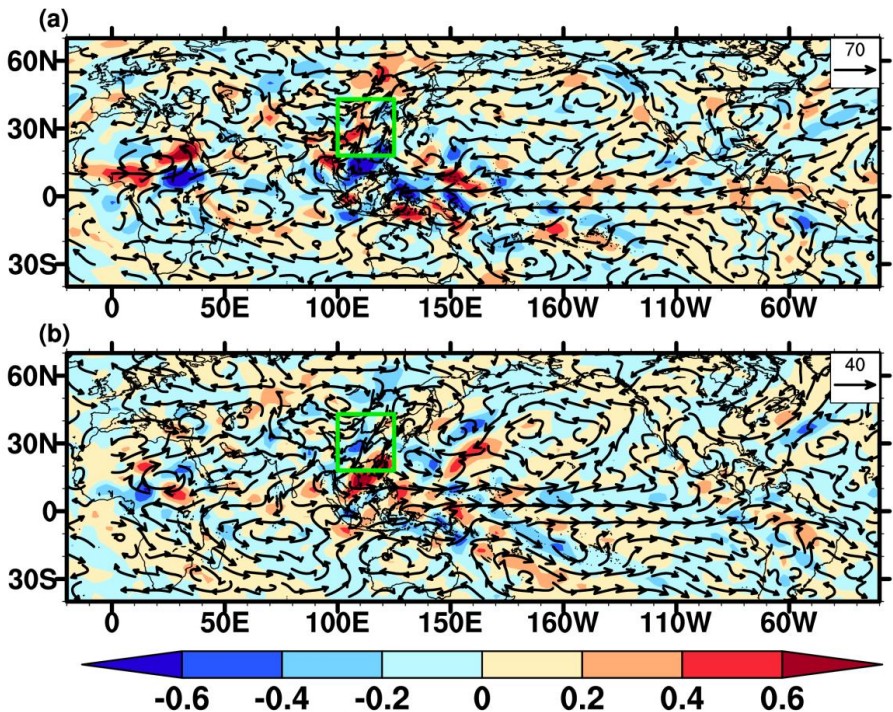

Figure 11. Vertically integrated water vapor anomalies (vector) and water vapor flux

divergence (shading) composited from the lightest 25th (a) and highest 75th (b)

percentile rainfall events in wet season. The water vapor flux unit is kg m$^{-1}$ s$^{-1}$ for and

the water vapor flux divergence is kg m$^{-2}$ s$^{-1}$. Green rectangle denotes the eastern

China.




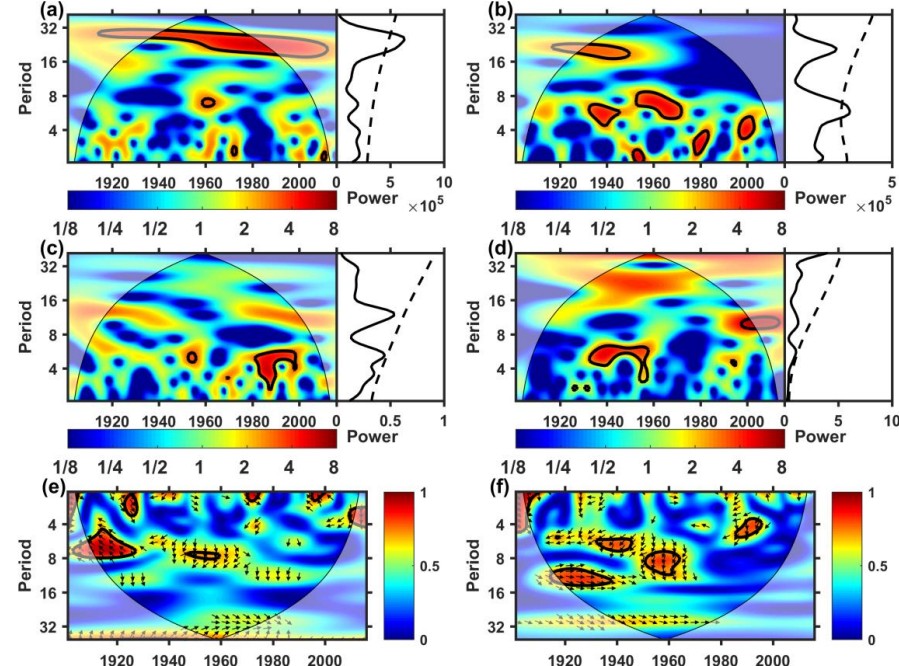


Figure 12. Wavelet spectra for wet season. (a) PC1, (b) PC2, (c) Niño3.4 index, (d)
PDO index, (e) wavelet spectral coherence of PC1 and Niño3.4, and (f) wavelet
spectral coherence of PC2 and PDO. The global spectra are shown on the right side of
the time varying wavelet spectra and, the black lines denote the statistical significance
at the 95% confidence level.




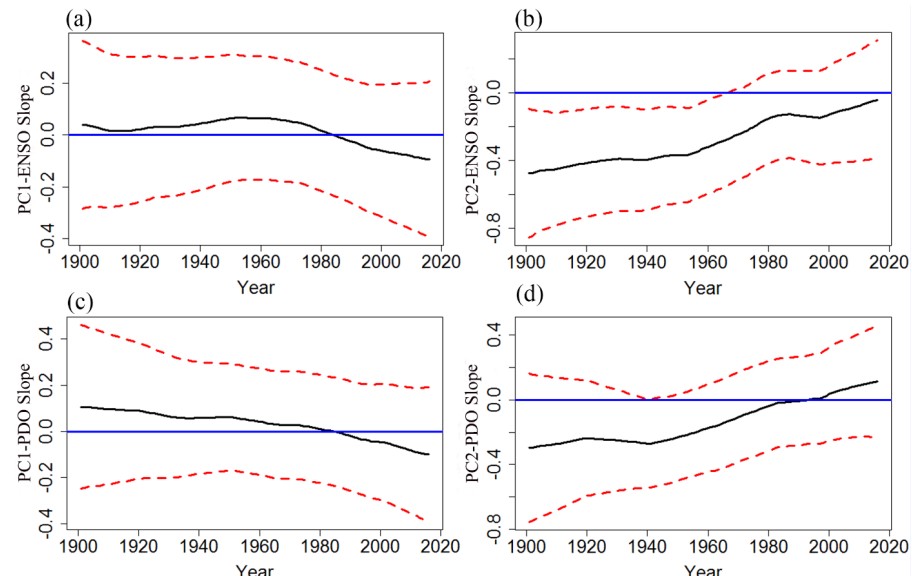

Figure 13. Changes in the relationships between rainfall and ENSO/PDO over time during 1901-2015. Black solid lines denote the estimated time-varying slopes, along with 25th and 75th percentile credible interval lines (red dashed lines) from the Bayesian dynamic linear model analysis.

855

856