# Peer review of "The precipitation variability of wet and dry season at the interannual and interdecadal scales over eastern China (1901–2016): The impacts of the Pacific Ocean"

_Hydrology and Earth System Sciences, 2020_

## Referee Comment (RC1) · Anonymous Referee #1 · 14 Apr 2020

This paper attempted to investigate the variability of precipitation in China using a long-term dataset and integrating multiple statistical methods such as PCA/EOF, wavelet analysis, and Bayesian dynamical linear model. However, the results presented in this paper are very well understood in the literature, and the impacts of ENSO and PDO have been studied extensively in existing studies (the authors also introduce some of them in their introduction section). The authors should have done a thorough investigation of the research gap, and the novelty of their current paper should be clearly stated. It seems that the authors thought utilizing April–September as the wet half year

(wet season) and October–March as the wet half-year is novel. That is not convincing.

Some similar studies are as follows (there are much more than that):

Ouyang, R., Liu, W., Fu, G., Liu, C., Hu, L., & Wang, H. (2014). Linkages between ENSO/PDO signals and precipitation, streamflow in China during the last 100 years. Hydrol. Earth Syst. Sci, 18(9), 3651-3661.

Yang, Q., Ma, Z., & Xu, B. (2017). Modulation of monthly precipitation patterns over East China by the Pacific Decadal Oscillation. Climatic change, 144(3), 405-417.

Yang, Q., Ma, Z., Fan, X., Yang, Z. L., Xu, Z., & Wu, P. (2017). Decadal modulation of precipitation patterns over eastern China by sea surface temperature anomalies. Journal of Climate, 30(17), 7017-7033.

Xiao, M., Zhang, Q., & Singh, V. P. (2015). Influences of ENSO, NAO, IOD and PDO on seasonal precipitation regimes in the Yangtze River basin, China. International Journal of Climatology, 35(12), 3556-3567.

—————————————————

---

## Referee Comment (RC2) · Anonymous Referee #2 · 10 Sep 2020

This paper aims to identify the dominant variability modes of precipitation in dry (October- March) and wet seasons over eastern China. Here are some suggestions to improve the manuscript. 1. It is suggested to highlight more clearly the novelty of this study in the Introduction Section. As in the current version, it cannot be clearly seen. 2. Before introducing different precipitation datasets and different analysis methods, it is suggested to briefly state how the datasets are related, and how using different datasets (methods) serve different analysis purposes. 3. There are some minor mistakes require attention, for instance, L215, symbol 'alpha' cannot be properly displayed.

[Figure]

L237∼238 and L791: grammar mistakes.
* * *

---

## Author Response (AR1)

September 26, 2020

Dear Prof. Dominic Mazvimavi,

Thank you very much for your letter and for the reviewers' constructive and valuable comments concerning our manuscript entitled "The precipitation variability of wet and dry season at the interannual and interdecadal scales over eastern China (1901–2016): The impacts of the Pacific Ocean" (Manuscript ID: hess-2020-102). In the revised version, we have carefully revised the paper to address all comments, and hope you and the reviewers find the revision satisfactory. Point-to-point responses to reviewers' comments are included in the following pages.

We look forward to hearing from you!

Sincerely yours

Fuqiang Cao E-mail: wq2006126@126.com

**Responses to the Comments of Reviewer 1**

This paper attempted to investigate the variability of precipitation in China using a longterm dataset and integrating multiple statistical methods such as PCA/EOF, wavelet analysis, and Bayesian dynamical linear model. However, the results presented in this paper are very well understood in the literature, and the impacts of ENSO and PDO have been studied extensively in existing studies (the authors also introduce some of them in their introduction section). The authors should have done a thorough investigation of the research gap, and the novelty of their current paper should be clearly stated. It seems that the authors thought utilizing April–September as the wet half year (wet season) and October–March as the wet half-year is novel. That is not convincing.

Some similar studies are as follows (there are much more than that):

*Ouyang, R., Liu, W., Fu, G., Liu, C., Hu, L., & Wang, H. (2014). Linkages between ENSO/PDO signals and precipitation, streamflow in China during the last 100 years. Hydrol. Earth Syst. Sci, 18(9), 3651-3661.*

Yang, Q., Ma, Z., & Xu, B. (2017). Modulation of monthly precipitation patterns over East China by the Pacific Decadal Oscillation. Climatic change, 144(3), 405-417.

Yang, Q., Ma, Z., Fan, X., Yang, Z. L., Xu, Z., & Wu, P. (2017). Decadal modulation of precipitation patterns over eastern China by sea surface temperature anomalies. Journal of Climate, 30(17), 7017-7033.

Xiao, M., Zhang, Q., & Singh, V. P. (2015). Influences of ENSO, NAO, IOD and PDO on seasonal precipitation regimes in the Yangtze River basin, China. International Journal of Climatology, 35(12), 3556-3567.

**Response:**

Thank you so much for your constructive comments. We have made a substantial revision of the paper to address all the issues.

The research gap and novelty have been highlighted in the revised manuscript as follows.

Most existing studies focusing on the effects of ENSO and PDO on precipitation over eastern China mainly examine the spatial pattern of rainfall during different phases of climate variability modes, while the time-varying linkages between eastern China rainfall and large-scale modes have not been investigated. However, predictability of seasonal rainfall over the East Asia largely depends on the phase and magnitude of the climate variability modes, as well as the relationship between large-scale modes and regional precipitation (Chan and Zhou, 2005; Wang et al., 2020), quantifying these corrections are, therefore, greatly instrumental to developing skillful precipitation forecasting model (Zhang et al., 2014). In this study, we used wavelet analysis and Bayesian dynamical linear model to analyze their time-varying relationships at the century-scale, this may fill the research gap of the century-scale time-varying linkages between climate variability modes and regional rainfall events.

We have added related descriptions and discussions in abstract (lines 41-46), and introduction (lines 132-157), as well as discussion and conclusions (lines 504-518).

Although the rainfall events mainly occur in summer (June-August), the rainy season extends April–September over eastern China, since the rainfall in eastern China is principally concentrated during April–September (Bao 1987; Domroes and Peng 1988; Zhai et al., 2005; Wang et al., 2020). Usage of boreal standard seasons may therefore unavoidably break the natural rainy distribution at the temporal scale, affecting the robustness of the analytical results. Zhai et al. (2005) have investigated trends of precipitation extremes during wet season (April–September) and dry season (October–March) in China, and suggested that utilization of six months as the dry (wet) half year facilitates to characterize the variations in extreme events. While up to now, the issue on whether the ENSO and PDO can contribute to the interannual and interdecadal rainfall variability in major rainy seasons over eastern China remains unclear. In this study, we utilize April–September as the wet half year (wet season) and October–March as the dry half year (dry season), respectively, to fill the gap of detecting robust signals of the time-varying effects of ENSO and PDO on the precipitation variability in eastern China based on long-term datasets.

These are also discussed in the revised manuscript.

**References**

- Bao, C. L. : Synoptic Meteorology in China. China Ocean Press, 209 pp, 1987.
- Chan, J. C., and Zhou, W.: PDO, ENSO and the early summer monsoon rainfall over south China, Geophys Res Lett, 32, L08810, https://doi.org/10.1029/2004GL022015, 2005.
- Domroes, M., and G. Peng: The Climate of China. SpringVerlag, 361 pp, 1988.
- Zhai, P., Zhang, X., Wan, H., and Pan, X.: Trends in total precipitation and frequency of daily precipitation extremes over China, *J Climate*, 18, 1096-1108, https://doi.org/10.1175/JCLI-3318.1, 2005.
- Zhang, W., Jin, F. F., and Turner, A.: Increasing autumn drought over southern China associated with ENSO regime shift, *Geophys Res Lett*, 41, 4020-4026, https://doi.org/10.1002/2014GL060130, 2014.
- Wang, B., Luo, X., and Liu, J.: How Robust is the Asian Precipitation–ENSO Relationship during the Industrial Warming Period (1901–2017)? J Climate, 33, https://doi.org/10.1175/JCLI-D-19-0630.1, 2779-2792, 2020.

**Responses to the Comments of Reviewer 2**

This paper aims to identify the dominant variability modes of precipitation in dry (October- March) and wet seasons over eastern China. Here are some suggestions to improve the manuscript.

**Response:** Thanks for your careful review and helpful comments. We have carefully revised our paper accordingly and believe that the quality of the revised paper has been greatly improved.

**Specific comments:**

1. It is suggested to highlight more clearly the novelty of this study in the Introduction Section. As in the current version, it cannot be clearly seen.

**Response:** Thank you very much for this suggestion. We have changed the section of introduction and highlighted the novelty in comparison with existing studies. For detailed information please see lines 62-97 and lines 132-157 in the revised version.

2. Before introducing different precipitation datasets and different analysis methods, it is suggested to briefly state how the datasets are related, and how using different datasets (methods) serve different analysis purposes.

**Response:** Thank you very much for your advice. We have added related descriptions of the relationships and utilizations of the rainfall dataset following your suggestion. For detailed information please see lines 174-181 in the revised version.

3. There are some minor mistakes require attention, for instance, L215, symbol 'alpha' cannot be properly displayed.

**Response:** Thank you so much for pointing out this. We have retyped this symbol in the standard format.

**4. L237 ~238 and L791: grammar mistakes.**

**Response:** Many thanks for this suggestion. We have changed this sentence as follows,

"These changes in rainfall confirm that it is reasonable to categorize wet and dry seasons over eastern China."

We deleted "for" in line 791.

In addition, we also have made a careful proofreading for the revised manuscript. Please see these minor changes in the revised version.

---

## Author Response (AR2)

February 13, 2021

Dear Prof. Dominic Mazvimavi,

Thank you very much for your letter and for the reviewers' constructive and valuable comments concerning our manuscript entitled "The precipitation variability of wet and dry season at the interannual and interdecadal scales over eastern China (1901–2016): The impacts of the Pacific Ocean" (Manuscript ID: hess-2020-102). In the revised version, we have carefully revised the paper to address all comments, and hope you and the reviewers find the revision satisfactory. Point-to-point responses to reviewers' comments are included in the following pages.

We look forward to hearing from you!

Sincerely yours

Fuqiang Cao

E-mail: wq2006126@126.com

**Responses to the Comments**

*Thanks for your encouragement and constructive comments. We have followed your suggestions and carefully revised the manuscript to address the issues raised by you.*

*1.  Line 43, In -> During*

**Response:**

Modified as suggested. Please see Line 43 on Page 2 in the revised manuscript.

*2.  Line 43, exerts -> exerted*

**Response:**

Modified as suggested. Please see Line 43 on Page 2 in the revised manuscript.

*3.  Line 44, decrease -> decreased*

**Response:**

Modified as suggested. Please see Line 44 on Page 2 in the revised manuscript.

*4.  Line 45, shift -> shifted*

**Response:**

Modified as suggested. Please see Line 45 on Page 2 in the revised manuscript.

*5.  Line 45, accessing -> assessing*

**Response:**

Modified as suggested. Please see Line 46 on Page 2 in the revised manuscript.

*6.  Line 48, "Precipitation over eastern China" —This cannot be a keyword. This is almost a sentence.*

**Response:**

We have divided this sentence into "Precipitation" and "Eastern China" according to your suggestion. Please see Line 48 on Page 2 in the revised manuscript.

*7. Line 53, derived from -> due to the*

**Response:**

Modified as suggested. Please see Line 53 on Page 2 in the revised manuscript.

*8. Line 68, delete "While"*

**Response:**

Modified as suggested. Please see Line 68 on Page 3 in the revised manuscript.

*9. Line 71, delete "Moreover"*

**Response:**

Modified as suggested. Please see Line 71 on Page 3 in the revised manuscript.

*10. Line 77, for -> of*

**Response:**

Modified as suggested. Please see Line 76 on Page 4 in the revised manuscript.

*11. Line 79, , it -> . It*

**Response:**

Modified as suggested. Please see Line 79 on Page 4 in the revised manuscript.

*12. Line 81, delete "the"*

**Response:**

Modified as suggested. Please see Line 81 on Page 4 in the revised manuscript.

*13. Line 90, delete "be"*

**Response:**

Modified as suggested. Please see Line 90 on Page 4 in the revised manuscript.

*14. Lines 90-91, Sentence not clear. Do you mean the amount of rainfall increases?*

*Or the relationship between amount and ENSO strengthens/weakens?*

**Response:**

Thank you for your suggestion. We have changed this sentence according to your advice as follows:

*"**Moreover, the summertime rainfall amount over the YRB and to its south is expected to increase (decrease) during El Niño (La Niña) years.**"*

*15. Line 95, delete "plausibly"*

**Response:**

Modified as suggested. Please see Line 95 on Page 5 in the revised manuscript.

*16. Lines 96-97, not very clear*

**Response:**

Thank you for your advice. We have changed this sentence according to your suggestion as follows:

*"**because the western Pacific warm pool shifts the WPSH northward**"*

*17. Line 102, dry and wet alternations -> the alternating of dry and wet years*

**Response:**

Modified as suggested. Please see Line 101 on Page 5 in the revised manuscript.

*18. Line 112, strengthened -> relatively high; weakened -> low*

**Response:**

Modified as suggested. Please see Line 111 on Page 5 in the revised manuscript.

*19. Line 119, also -> been*

**Response:**

Modified as suggested. Please see Line 118 on Page 6 in the revised manuscript.

*20. Line 124, the -> a*

**Response:**

Modified as suggested. Please see Line 123 on Page 6 in the revised manuscript.

*21. Line 143, not clearly written. are you referring to the variability of the seasonality of precipitation? how did you define seasonality for analytical purposes?*

**Response:**

Thank you very much for your good suggestion. We have changed these sentences according to your advice as follows:

***"Moreover, the variations in climatological seasonal rainfall are employed in aforementioned analyses, while the main rainy season in China, in particularly for eastern China, does not follow conventional seasonal boundaries, since the rainfall in eastern China is principally concentrated during April–September"***

*22. Line 150, to characterize the -> characterisation of*

**Response:**

Modified as suggested. Please see Line 148 on Page 7 in the revised manuscript.

*23. Line 151, delete "While up to now, the issue on whether"*

**Response:**

Modified as suggested. Please see Line 149 on Page 7 in the revised manuscript.

*24. Line 151, the -> The contribution of both*

**Response:**

Modified as suggested. Please see Line 149 on Page 7 in the revised manuscript.

*25. Line 152, delete "can contribute"*

**Response:**

Modified as suggested. Please see Line 149 on Page 7 in the revised manuscript.

*26. Line 153, utilize -> consider*

**Response:**

Modified as suggested. Please see Line 151 on Page 7 in the revised manuscript.

*27. Line 155, to fill the gap of detecting robust signals of the -> to examine the*

**Response:**

Modified as suggested. Please see Line 152 on Page 7 in the revised manuscript.

*28. Line 162, developed at -> managed by the*

**Response:**

Modified as suggested. Please see Line 159 on Page 7 in the revised manuscript.

*29. Line 165, delete "The accurate"*

**Response:**

Modified as suggested. Please see Line 162 on Page 8 in the revised manuscript.

*30. Lines 166-170, Did the author do this quality control or this was done by the National Meteorological Centre?*

**Response:**

Thank you for your suggestion. We conduct these works of quality control, and we have changed these sentences according to your advice as follows:

*"We conduct the accurate quality control procedures to check the temporal inhomogeneity and missing values, and screen the related stations in the following analyses"*

*31. Line 175, more -> Further; referred to -> available in*

**Response:**

Modified as suggested. Please see Line 172 on Page 8 in the revised manuscript.

*32. Line 192, delete "the"*

**Response:**

Modified as suggested. Please see Line 189 on Page 9 in the revised manuscript.

*33. Line 197, The -> A; delete "the"*

**Response:**

Modified as suggested. Please see Line 194 on Page 9 in the revised manuscript.

*34. Line 198, refers to -> is available in*

**Response:**

Modified as suggested. Please see Lines 194-195 on Page 9 in the revised manuscript.

*35. Line 199, are you referring to temporal or spatial changes?*

**Response:**

Thank you for your advice. We have changed these sentences according to your suggestion as follows:

***"To identify the effects of climate variability modes on spatio-temporal changes in rainfall over eastern China"***

*36. Line 201, which variables are these?*

**Response:**

Thank you for your suggestion. We have changed these sentences according to your advice as follows:

***"variables (e.g., sea surface temperature, sea level pressure and vertically integrated water vapor)"***

*37. Line 206, access -> assess*

**Response:**

Modified as suggested. Please see Line 205 on Page 10 in the revised manuscript.

*38. Lines 209-235, All symbols and letters used in equations should be defined. Avoid having the same letter representing different variables, constants or coefficients in different parts of the paper, e.g. y and x.*

**Response:**

Thank you so much for the good suggestion. We have changed these according to your advices. Please see Lines 208-205 on Pages 10-11 in the revised manuscript.

*39. Line 264, Not clear what you are referring to*

We have changed these sentences according to your suggestion as follows:

*"In addition, the spectral analysis is also performed using the time series of mean rainfall based on the two datasets"*

*40. Line 273, correct this phrase. Not clear.*

We have changed these sentences according to your suggestion as follows:

*"which is well consistent with the spatial mean rainfall"*

*41. Line 335, add "The"*

**Response:**

Modified as suggested. Please see Line 336 on Page 16 in the revised manuscript.

*42. Lines 351-352, Not clear*

We have changed these sentences according to your suggestion as follows:

*"The solid line indicates the decadal smoothing average of first PC"*

*43. Line 358, add "a"*

**Response:**

Modified as suggested. Please see Line 359 on Page 17 in the revised manuscript.

*44. Lines 360-361, Sentence incomplete*

Thank you for your suggestion. We have changed these sentences according to your advice as follows:

*"early summer rainy season presents in southern China, then the mei-yu season occurs over the Yangtze-Huai river basins, and the late summer rainy season*

*ultimately forms over northern China"*

*45. Line 379, , this -> . This*

**Response:**

Modified as suggested. Please see Line 380 on Page 18 in the revised manuscript.

*46. Line 397, delete "lightest"*

**Response:**

Modified as suggested. Please see Line 398 on Page 18 in the revised manuscript.

*47. Line 403, delete "highest"*

**Response:**

Modified as suggested. Please see Line 404 on Page 19 in the revised manuscript.

*48. Line 406, suitably -> conducive*

**Response:**

Modified as suggested. Please see Line 407 on Page 19 in the revised manuscript.

*49. Line 408, light -> low*

**Response:**

Modified as suggested. Please see Line 409 on Page 19 in the revised manuscript.

*50. Line 418, season -> seasons*

**Response:**

Modified as suggested. Please see Line 419 on Page 19 in the revised manuscript.

*51. Line 430, accessed -> assessed*

**Response:**

Modified as suggested. Please see Line 431 on Page 20 in the revised manuscript.

*52. Line 446, are -> have*

**Response:**

Modified as suggested. Please see Line 447 on Page 21 in the revised manuscript.

*53. Line 460, season -> seasons*

**Response:**

Modified as suggested. Please see Line 461 on Page 21 in the revised manuscript.

*54. Line 467, season -> seasons*

**Response:**

Modified as suggested. Please see Line 468 on Page 22 in the revised manuscript.

*55. Line 474, delete "occurred"; study -> the study*

**Response:**

Modified as suggested. Please see Line 475 on Page 22 in the revised manuscript.

*56. Line 480, season -> seasons*

**Response:**

Modified as suggested. Please see Line 481 on Page 22 in the revised manuscript.

*57. Line 481, with that during dry season -> from the dry seasons; the lightest -> low*

**Response:**

Modified as suggested. Please see Lines 481-482 on Page 22 in the revised manuscript.

*58. Line 489, to -> in*

**Response:**

Modified as suggested. Please see Line 489 on Page 23 in the revised manuscript.

*59. Line 511, These -> The results*

**Response:**

Modified as suggested. Please see Line 511 on Page 24 in the revised manuscript.

*60. Line 541, heartily -> very*

**Response:**

Modified as suggested. Please see Line 541 on Page 25 in the revised manuscript.

*61. Line 800, delete "climatological"*

**Response:**

Modified as suggested. Please see Line 800 on Page 37 in the revised manuscript.

*62. Line 815, in -> by the*

**Response:**

Modified as suggested. Please see Line 815 on Page 39 in the revised manuscript.